# Ventilator-Associated Pneumonia due to Drug-Resistant *Acinetobacter baumannii*: Risk Factors and Mortality Relation with Resistance Profiles, and Independent Predictors of In-Hospital Mortality

**DOI:** 10.3390/medicina55020049

**Published:** 2019-02-13

**Authors:** Aušra Čiginskienė, Asta Dambrauskienė, Jordi Rello, Dalia Adukauskienė

**Affiliations:** 1Department of Intensive Care, Medical Academy, Lithuanian University of Health Sciences, 50161 Kaunas, Lithuania; 2Infection Control Service, Medical Academy, Lithuanian University of Health Science, 50161 Kaunas, Lithuania; dambrauskienea@gmail.com; 3Department of Intensive Care, Vall d’Hebron University Hospital, 08035 Barcelona, Spain; jrello@crips.es; 4CIBERES, 08035 Barcelona, Spain; 5Universitat Autonoma de Barcelona, 08193 Barcelona, Spain; 6Vall d’Hebron Institute of Research, 08035 Barcelona, Spain; 7CRIPS, Vall d’Hebron Institute of Research (VHIR), 08035 Barcelona, Spain

**Keywords:** *A. baumannii*, ventilator-associated pneumonia (VAP), drug resistance, mortality, predictors

## Abstract

*Background and objectives:* High mortality and healthcare costs area associated with ventilator-associated pneumonia (VAP) due to *Acinetobacter baumannii* (*A. baumannii*). The data concerning the link between multidrug-resistance of *A. baumannii* strains and outcomes remains controversial. Therefore, we aimed to identify the relation of risk factors for ventilator-associated pneumonia (VAP) and mortality with the drug resistance profiles of *Acinetobacter baumannii* (*A. baumannii*) and independent predictors of in-hospital mortality. *Methods:* A retrospective ongoing cohort study of 60 patients that were treated for VAP due to drug-resistant *A. baumannii* in medical-surgical intensive care units (ICU) over a two-year period was conducted. *Results:* The proportions of multidrug-resistant (MDR), extensively drug-resistant (XDR), and potentially pandrug-resistant (pPDR) *A. baumannii* were 13.3%, 68.3%, and 18.3%, respectively. The SAPS II scores on ICU admission were 42.6, 48.7, and 49 (*p* = 0.048); hospital length of stay (LOS) prior to ICU was 0, one, and two days (*p* = 0.036), prior to mechanical ventilation (MV)—0, 0, and three days (*p* = 0.013), and carbapenem use prior to VAP—50%, 29.3%, and 18.2% (*p* = 0.036), respectively. The overall in-hospital mortality rate was 63.3%. In MDR, XDR, and pPDR *A. baumannii* VAP groups, it was 62.5%, 61.3%, and 72.7% (*p* = 0.772), respectively. Binary logistic regression analysis showed that female gender (95% OR 5.26; CI: 1.21–22.83), SOFA score on ICU admission (95% OR 1.28; CI: 1.06–1.53), and RBC transfusion (95% OR 5.98; CI: 1.41–25.27) were all independent predictors of in-hospital mortality. *Conclusions:* The VAP risk factors: higher SAPS II score, increased hospital LOS prior to ICU, and MV were related to the higher resistance profile of *A. baumannii*. Carbapenem use was found to be associated with the risk of MDR *A. baumannii* VAP. Mortality due to drug-resistant *A. baumannii* VAP was high, but it was not associated with the *A. baumannii* resistance profile. Female gender, SOFA score, and RBC transfusion were found to be independent predictors of in-hospital mortality.

## 1. Introduction

Ventilator-associated pneumonia (VAP) is one of the most common intensive care unit (ICU)-acquired infections that are associated with a prolonged duration of antibacterial treatment, length of hospital stay (LOS), and mechanical ventilation (MV), as well as high mortality and healthcare costs [1,2,3,4]. *Acinetobacter baumannii* (*A. baumannii*) is one of the most prevalent VAP-causing pathogens [3,5,6]. The incidence of VAP due to Acinetobacter spp. varies across different goegraphic regions [7]. Although infections due to Acinetobacter spp. were originally thought to be associated with humid climates, in recent decades, this infection has also rapidly spread in temperate climates [8].

*Acinetobacter* is an obligate aerobic nonfermenting gram-negative nonmotile bacterium that was discovered by Dutch microbiologist Martinus Willem Beigerinck in 1911 [9]. For a long time, Acinetobacter was considered to be bacteria of low virulence, being susceptible to commonly used antibacterial agents, but since the 1970s its resistance began to increase and then became a serious problem, especially in nosocomial settings [4,9]. Nowadays, infections that are due to *A. baumannii* are recognized to be one of the most threatening and are difficult to control and to treat in critical care settings [3,4,5,10,11,12]. *A. baumannii* survives equally in both dry and humid environments, is resistant to disinfectants and ultimate drying, and is able to form biofilms that facilitate bacterial bonding to tissues, also various environmental surfaces, devices, and quickly acquiring various antibiotic resistance mechanisms [8]. It is believed that these properties have led to the rapid endemic spread of *A. baumannii* in the hospital environment and many ICUs worldwide, particularly within Europe. In 2015, the European antimicrobial resistance surveillance network report stated that the share of drug-resistant *A. baumannii* strains across Europe was steadily increasing [13]. The highest levels of *A. baumannii* drug resistance were observed in Southern and Southeastern Europe and in the Baltic States, especially within Lithuania. In 2017, *A. baumannii* was included in the WHO global priority list of drug-resistant bacteria in order to highlight the need for research development and the urgency for new antibiotics [14]. According to the data of the Lithuanian Center for Communicable Diseases and AIDS, the rate of VAP in Lithuania has increased from 15.9 to 30.3 cases per 1000 mechanical ventilation days in the period of 2014–2017 [15,16]. *A. baumannii* was identified as the most common causative agent of VAP [15]. The drug resistance of *A. baumannii* strains increased by 21% in carbapenems and aminoglycosides, and by 25.9% in fluoroquinolones in the period of 2014–2017 in Lithuania [17].

Treatment of infections due to *A. baumannii* remains challenging. High rates of native resistance, as well as the rapid increase of acquired resistance to commonly prescribed antibiotics classes limits the choice of active antibacterial treatment and compromises the course of disease and patient’s outcomes. Historically, imipenem therapy was the “gold standard for pneumonia due to *A. baumannii* [18]. Later, the selection of empirical treatment was recommended, depending on the time of onset of VAP and the presence of risk factors for resistant microorganisms [1]. For patients with late-onset disease or risk factors for multidrug-resistant pathogen combination antibiotic therapy (antipseudomonal cephalosporin or carbapenem or ß-lactam/ß-lactamase inhibitor plus antipseudomonal fluoroquinolone or aminoglycoside) was recommended [1]. Due to the progress of antibacterial resistance, alternative options were scant; therefore, it has led to a return to the old, less effective, and more toxic polymyxins and tetracyclines, for which this bacterium still remains sensitive. Often, colistin is the last-resort medicine, despite nephrotoxicity and neurotoxicity; poor penetration to the lung tissue also continues to limit its’ use in VAP treatment. Moreover, resistance to the last-choice colistin has also gained a threatening spread in recent years [4]. Therefore, if high drug resistance of *A. baumannii* is prevalent in a hospital or departments, empirical combined treatment with colistin plus carbapenem, sulbactam, or tigecycline should be used.

Greater mortality and higher health care associated costs were found to be associated with the delayed recognition and treatment of VAP due to drug-resistant *A. baumannii* [4]. Age, previous hospitalizations, surgery, invasive monitoring and treatment procedures, and comorbidities were identified as risk factors for VAP [1]. Hospital LOS, previous antibacterial treatment, duration of MV, disease severity, and prevalence of drug-resistant *A. baumannii* strains in hospitals in the community have also been recognized as risk factors namely for VAP due to multidrug-resistant *A. baumannii* [1,2,19]. Unfortunately, there is no data regarding the relation between *A. baumannii* resistance profiles (multidrug-resistant (MDR), extensively drug-resistant (XDR), and potentially pandrug-resistant (pPDR)), and the risk factors of VAP as well as the patient outcomes in Lithuania that are based on microbiological and clinical researches. Knowledge about the risk factors and mortality association with different degrees of antibacterial resistance of *A. baumannii*, and the estimation of mortality predictors of these patients are relevant for more rapid diagnosis, urgent treatment, and they may influence the improvement of patients’ outcomes. It is unclear whether the results of the studies that were conducted in other countries can be extrapolated to Lithuanian patients due to the differences in regional healthcare systems as well as prevailing VAP pathogens and their mechanisms of drug-resistance. The aim of our study was specifically to identify the relation of risk factors for ventilator-associated pneumonia (VAP) and mortality with the resistance profiles of *A. baumannii*, while also estimating the independent predictors of in-hospital mortality.

## 2. Materials and Methods

A retrospective cohort study of medical records of patients that were admitted to 18-bed medical-surgical adult ICUs was conducted at Lithuania’s largest 2300-bed university-affiliated hospital over a two-year period (from January 2014 to December 2015). Kaunas Regional Biomedical Research Ethics Committee approved this study (No. BE-2-13). The need for written consent was waived due to the retrospective nature of the study.

Inclusion criteria were as follows: (1) age ≥ 18 year, (2) the first episode of VAP due to drug-resistant *A. baumannii*. Pneumonia was considered to be ventilator-associated when it occurred after more than 48 h after the onset of mechanical ventilation. Clinical diagnosis of VAP was made according to 2005 ATS/IDSA criteria [1]. Sepsis and septic shock were diagnosed according to Sepsis-2 criteria [20].

Data collected for each VAP case included age, gender, type of admission (internal disease/surgical/trauma), hospital, and ICU LOS, MV prior to and after VAP diagnosis, and antibacterial treatment days prior to VAP diagnosis, the presence of chronic illness (diabetes mellitus, heart, neurologic and obstructive pulmonary disease, renal and hepatic failure, and malignancy), sepsis status, drug resistance of *A. baumannii* isolates, history of antibiotics used, red blood cell (RBC) transfusion, surgery, reintubation, tracheostomy, coma, and outcome (discharge or death). Mortality included all deaths occurring during the hospital stay among patients with VAP due to drug-resistant *A. baumannii*. Severity of illness was assessed on ICU admission while using the Sequential Organ Failure Assessment (SOFA), Simplified Acute Physiology Score (SAPS) II, and Acute Physiology and Chronic Health Evaluation (APACHE) II scores.

The identification of *A. baumannii* isolates and antibiotic susceptibility was performed according to the EUCAST guidelines [21]. Drug-resistance profiles of *A. baumannii* were defined as MDR, XDR, or potentially PDR (pPDR) according to an international expert proposal for the interim standard definitions for acquired resistance criteria [22]. We decided to use the pPDR category of *A. baumannii* resistance instead of PDR due to the incomplete testing of drug resistance in our hospital that was performed until summer 2015 (testing to colistin was not used).

### Statistics

The variables were summarized as frequencies and percentage, means and standard deviation, or medians and interquartile ranges. The normality of data was tested using the Shapiro–Wilk normality test. Student’s *t* test, Mann–Whitney non-parametric test, Pearson’s chi-square test, two-tailed Fisher’s exact test, or Kruskal–Wallis test were performed to detect the differences between groups as appropriate. All of the variables that were measured were subjected to univariate analyses. Binary logistic regression analysis was applied to identify the independent predictors of in-hospital mortality. A receiver operating characteristic (ROC) analysis was used to determine the area under the curve (AUC) and the cut-off values of SAPS II, APACHE II, and SOFA scores for predicting mortality. In all analyses, two-sided *p* values of < 0.05 were considered to be statistically significant. Statistical analysis was carried out while using the Statistical Package for the Social Sciences SPSS. version 20 (SPSS, Chicago, IL, USA).

## 3. Results

The data of 73 mechanically ventilated medical-surgical ICU patients with tracheal aspirates that were positive for drug-resistant *A. baumannii* in the Hospital of Lithuanian University of Health Sciences Kauno Klinikos during the two-year period were analyzed. A total of 13 cases did not meet the diagnostic criteria for VAP; therefore, 60 cases were included in the final study. There were 29 (48.3%) women and 31 (51.7%) men with a mean age of 64 (SD 15) years.

### 3.1. Associations between the VAP Pathogen A.baumannii Drug Resistance Profile and Risk Factors for VAP

MDR, XDR, and pPDR *A. baumannii* strains caused 13.3%, 68.3%, and 18.3% of all cases regarding drug-resistant *A. baumannii* VAP, respectively (*p* < 0.05). The relationship between the *A. baumannii* drug resistance profiles and the risk factors for VAP is presented in Table 1.

The groups of MDR, XDR, and pPDR *A. baumannii* were homogeneous according to age, gender, and type of admission (*p* > 0.05). The majority of patients were admitted due to internal diseases in all three drug resistance groups: 36.4%—due to pulmonary, 15.2%—due to gastrointestinal, 12.1%—due to kidney diseases, 12.1%—due to intoxication, and 25.5%—due to miscellaneous internal diseases. The differences were found in the SAPS II score on ICU admission: the score was the highest in the pPDR (49, SD 13.5) and the lowest in MDR (42.6, SD 13.8) *A. baumannii* group (*p* = 0.048). The hospital LOS prior to ICU among different drug resistance groups was statistically significantly different as well. It was the longest in pPDR and the shortest MDR *A. baumannii* patients (2 days, IQR 0–10 vs. 0 days, IQR 0–0.75; *p* = 0.036). The longest hospital LOS before MV was detected in the pPDR *A. baumannii* group (3 days, IQR 0–6, *p* = 0.013). Rates of coma, sepsis, septic shock, chronic diseases, surgery prior to ICU, tracheostomy, reintubation, hospital LOS and ICU LOS, as well as the duration of MV prior to VAP development did not differ significantly among the different drug resistance profiles of *A. baumannii* (*p* > 0.05).

No statistically significant association between the duration of antibacterial treatment prior to VAP and the drug resistance profile of *A. baumannii* was detected (*p* = 0.475). The possible relationship between the MDR, XDR, and pPDR *A. baumannii* strains and the class of antibiotics used that was prior to VAP was analyzed. The antibiotics that were used most frequently in all three groups were cephalosporins and penicillins with or without beta-lactamase inhibitors (BLI). The MDR *A. baumannii* group received carbapenems more frequently when compared to the XDR and pPDR groups (50%, 29.3%, and 18.2%, respectively; *p* = 0.039).

### 3.2. The Relationship between A.baumannii Drug Resistance Profile and Mortality

The in-hospital mortality rate in patients with VAP that was caused by drug-resistant *A. baumannii* was 63.3%. The detailed data on in-hospital mortality in MDR, XDR, and pPDR *A. baumannii* groups are presented in Figure 1.

### 3.3. Predictors of In-Hospital Mortality

Univariate and multivariate analyses were performed in order to identify the independent predictors of in-hospital mortality. The in-hospital mortality rate was significantly higher in females than males (79.3% vs. 48.4%, *p* < 0.05). The disease severity on ICU admission based on SAPS II (51.5, SD 10.1) and SOFA (12.7, SD 3.8) scores was also associated with adverse outcomes (*p* < 0.05). The higher APACHE II score was observed in non-survivors as compared to survivors, but the difference was not statistically significant (21.63, SD 9.1 vs. 17.6, SD 7.1, *p* = 0.082). Non-survivors had chronic diseases (72.2% vs. 45%, *p* = 0.038) and RBC transfusions (75% vs. 25%, *p* = 0.03) more frequently. The remaining analyzed factors did not differ between the outcome groups. The univariate analyses of demographic, clinical, and treatment factors are shown in Table 2.

Sensitivity and specificity analyses (ROC curves) were performed in order to assess the ability of severity scores (SOFA, SAPS II, APACHE II) to predict in-hospital mortality in drug-resistant *A. baumannii* VAP patients. The discriminating ability and cut-off of the scores were as follows: AUC 0.73 (95% CI: 0.59–0.86), sensitivity 63.2%, specificity 72.7%, cut-off value 11.5 for SOFA (*p* = 0.004), AUC 0.68 (95% CI: 0.54–0.84), sensitivity 68.4%, specificity 63.6%, cut off value 46 for SAPS II (*p* = 0.016), and AUC 0.62 (95% CI: 0.54–0.84), sensitivity 68.4%, specificity 50%, and cut-off value 16.5 for APACHE II (*p* = 0.118). However, no statistically significant difference was found in the discriminative power among the scores (*p* > 0.05). The details are presented in Figure 2.

The stepwise backward binary logistic regression was used to determine the independent predictors of in-hospital mortality. The following variables were included: gender, disease severity score on ICU admission (SOFA, SAPS II scores), presence of chronic diseases, and RBC transfusion prior to the diagnosis of VAP. The independent predictors of in-hospital mortality in VAP due to drug-resistant *A. baumannii* patients were female gender, SOFA score on ICU admission, and the RBC transfusion prior to VAP. The details are presented in Table 3.

## 4. Discussion

VAP risk factors: disease severity on admission to ICU (SAPS II score), longer hospital LOS prior to ICU, and prior to MV have been associated with higher drug resistance of *A. baumannii*. The previous use of carbapenems was associated with the risk of the development of MDR *A. baumannii* VAP. The overall in-hospital mortality rate was high (63.3%), but the influence of *A. baumannii* drug resistance profile on mortality was not significant. The female sex, the SOFA score on ICU admission, and the need for RBC transfusion were found to be independent predictors for mortality.

The MDR, XDR, and pPDR *A. baumannii* VAP patient groups were homogenous for demographic, anamnestic, and clinical data on the admission to ICU. According to previous studies, the main risk factors for VAP due to *A. baumannii* were antibiotic therapy (especially of the broad spectrum), MV, hospital LOS, invasive procedures, disease severity, and the presence of chronic diseases [2,3,19]. In Inchai et al. [19] study, carbapenem treatment was associated with the risk of VAP due to all three types of *A. baumannii* drug-resistance profiles, in particular PDR, and odds ratio for MDR, XDR, and PDR *A. baumannii* VAP were 5.2, 6.3, and 12.84, respectively. Carbapenems use was also found to be a risk factor for XDR *A. baumannii* VAP in Li et al. study [3]. The risk factor for VAP due to PDR *A. baumannii* was previous colistin treatment as well [19]. We observed more frequent carbapenem use prior to VAP in MDR *A. baumannii* group (*p* = 0.036), as in Inchai et al. study [19], missing the association with XDR and pPDR due to an insufficient study population. We only investigated the relation between the drug resistance profile of *A. baumannii* as a VAP pathogen and the established risk factors for VAP. Although it would be worthwhile to identify the risk factors for VAP due to *A. baumannii* of separate drug resistance profiles by logistic regression analysis; however, significant prevailing of XDR strains came out with too small sample sizes in MDR and pPDR profiles to allow for this analysis to be performed.

Another risk factor for drug-resistant *A. baumannii* VAP is disease severity estimated by SAPS II, APACHE II, and SOFA scores [2,3,19]. More severely ill patients tend to develop VAP due to the more resistant (XDR or PDR) *A. baumannii* strain [23]. Evidently, incoherence in opinion exists regarding the relation of predictive values of severity scores with the drug resistance profile of *A. baumannii*. Inchai et al. [19] found that a higher SOFA score predicted XDR and the higher SAPS II score—PDR *A. baumannii* VAP. In the study by Ozgur et al. [2], a higher SAPS II score predicted the risk of XDR-*A. baumannii* VAP; however, Li et al. [3] demonstrated that the APACHE II score had the best predictive ability. According to our results, only the SAPS II score was found to be statistically significantly associated with the higher drug resistance of the *A. baumannii* strain (*p* = 0.048).

The ICU LOS prior to MV, prior to VAP, and hospital LOS prior to VAP did not differ among the *A. baumannii* drug resistance groups. However, we found that patients in the pPDR *A. baumannii* group in comparison with the MDR and XDR groups had longer hospital LOS prior to ICU (*p* = 0.036) and prior to MV (*p* = 0.013). Unlike our study, Ozgur et al. [2] indicated that longer hospital LOS was a statistically significant risk factor of XDR *A. baumannii* VAP. We did not analyze the relation of the hospital and the ICU LOS with different *A. baumannii* drug resistance profiles, as we think that, methodologically, it would be more accurate to investigate LOS prior to VAP as a possible risk factor for VAP. From our point of view, the LOS after the diagnosis of VAP as a component of the general hospital LOS is more reflective of the effectiveness of treatment and patient outcome. In addition, it may be related not only to the disease severity, but also to the prevailing drug resistance mechanisms in different hospitals or departments.

Finally, no differences among the various *A. baumannii* drug resistance profiles with regard to the duration of antibiotic therapy prior to VAP, the type of admission, sepsis, coma, tracheostomy, surgery, RBC transfusion, and reintubation rates were detected. In a sample case control study by Li et al. that was smaller than ours [3], COPD and heart disease were identified as risk factors for XDR *A. baumannii* VAP (*p* < 0.05). Unlike Li et al. [3], we only analyzed the presence of chronic diseases, but not isolated pathologies, and we did not identify the relation of it in general (*p* = 0.508).

Eventually, an in-hospital mortality rate of VAP due to *A. baumannii* is high, and according to Ozgur et al. [2] study can reach up to 85.3% in the cases of XDR *A. baumannii* VAP. However, the pathogen drug resistance has not been proven to influence mortality, because of this study did not clarify what the resistance profiles the non-XDR group consist of. The in-hospital mortality in our study was found to be 63.3%, which is comparable with other studies [23,24]. Actually, we expected in-hospital mortality to increase with the higher drug resistance of *A. baumannii*. Despite the higher mortality rate in pPDR *A. baumannii* VAP group (72.7%), this difference was not statistically significant (*p* = 0.77). Our results are consistent with those of Ozgur et al. [2], Li et al. [3], and Inchai et al. [23], where the higher mortality rate in more drug-resistant *A. baumannii* VAP patients also was not statistically significant. The current opinion states that the higher mortality rate in VAP due to drug-resistant pathogens is associated with concomitant illnesses and the disease severity in the presence of VAP, but not with the only VAP itself or specifically the drug resistance of *A. baumannii*.

Age, concomitant diseases, invasive treatment or monitoring methods, inappropriate antibacterial therapy, septic shock, and the severity of illness are proven as the prognostic factors for drug-resistant *A. baumannii* VAP patients’ mortality [2,3,5,23]. We could not analyze the mortality factors for each separate *A. baumannii* drug resistance profile due to an insufficient number of cases in the MDR and pPDR groups; therefore, we have only investigated the whole study population. So far, we have found the female gender, SOFA score on admission to ICU, and RBC transfusion to be independent predictors for mortality in VAP, due drug-resistant *A. baumannii*.

Currently, in addition to identifying individual prognostic factors, the prognostic accuracy of various prognostic models is being investigated. The data on the predictive ability of prognostic scores remain controversial. One of the most recent meta-analysis by Larsson et al. [25] examined the accuracy of the seven mortality predictive models (APACHE II, CPIS, Immunodeficiency, IBMP-10, VAP PIRO, SOFA, SAPS II, and APACHE III), which have been used in patients with VAP. All of the models had almost equal pooled predictive ability, with the range of AUC being from 0.64 to 0.72. APACHE II, SAPS II, and SOFA were used most frequently in the included studies. APACHE II showed the highest overall predictive ability (AUC 0.72). However, the studies that were included in the meta-analysis did not distinguish the subgroup of patients with VAP due to drug-resistant *A. baumannii*; therefore, it is unclear whether the results could be applied to them. Our results differ from those of Larsson et al. [25], as the APACHE II score on ICU admission did not differ significantly between the survivors and non-survivors in our study, but comprehensibly the SOFA and SAPS II scores were statistically significantly higher in the non-survivors group (*p* < 0.05). In addition, the higher SOFA score on ICU admission predicted in-hospital mortality in the multivariate analysis (OR 1.28, 95% CI: 1.06–1.53). Although the SOFA score was the least sensitive (63.2%) when compared to APACHE II and SAPS II, it has had the highest specificity (72.7%). Based on the AUC of the ROC curve, the discriminatory ability of the SOFA score was good (AUC 0.73). In contrast to Larson et al. meta-analysis [25], both APACHE II and SAPS II had poor discriminatory ability (AUC 0.62 and 0.68, respectively). Likewise to our study, Karakuzu et al. [26] also noticed a better discriminative ability of the SOFA score when compared to the APACHE II (AUC 0.82 vs. 0.62), but for *A. baumannii*, there was just less than the half of the examined pathogens and their resistance was not specified, as our study did, which eventually confirms the strongest significance of our study findings. Comparably, Inchai et al. [23] also found that the SOFA score on the VAP diagnosis day and, in contrast to our study, by the SAPS II score, statistically significantly predicted the XDR *A. baumannii* VAP patient mortality. In our opinion, the SAPS II score is also related to demographic and anamnesis data, which are supporting less correlation in the severity of VAP than clinical presentation in SOFA score. The relation between the APACHE II score and the drug-resistant *A. baumannii* VAP patients’ mortality rate was also confirmed by Tsioutis et al. [27] study, which differs from Karakuzu et al. [26], Inchai et al. [23], and our study. This is why conclusion regarding the accuracy of some scores has been not estimated yet.

In literature, data on gender interrelation with mortality varies: it is indicated that either mortality is higher in men, or gender has no significant influence in drug-resistant *A. baumannii* VAP patients [23,24], but in our study, female sex has a significantly higher death rate than men (79.3% vs. 48.2%, *p* = 0.017); also, it was an independent predictor for in-hospital mortality (OR 5.26, 95% CI: 1.21-22.83). Subsequently, in order to explain this gender and mortality interrelation, we compared the characteristics of the men and women subgroups. Women have been older than men (66.8 vs. 61.2 years, *p* = 0.035), and they more often have had at least one chronic disease (82.8% vs. 61.3%, *p* = 0.065). Similar to our results, Tsioutis et al. [27] has found higher mortality rate in females too, but only in a univariate analysis.

We found that RBC transfusion predicted in-hospital mortality in patients with drug-resistant *A. baumannii* VAP. Currently, no research has been performed regarding the linkage between the RBC transfusion and mortality in VAP patients. Data on mortality in other patient populations are controversial. In the systematic review of 46 observational studies, which has investigated the impact of blood transfusions on outcome in heterogeneous patient groups (trauma, general, cardiac and neurosurgery, orthopedic, cardiac, and general ICU), Marik et al. [28] found that RBC transfusion independently predicted death (pooled OR from 12 studies 1.7; 95% CI: 1.4–1.9), infectious complications (pooled OR from nine studies 1.8; 95% CI: 1.5–2.2), and ARDS (pooled OR from six studies 2.5; 95% CI: 1.6–3.3). In ICU patients, the three trials that were included in the review confirmed a statistically significant association between RBC transfusion and mortality. However, the SOAP study that was conducted in 198 European ICUs not only failed to detect association between RBC transfusion and higher mortality but it found that patients who received transfusion had a better survival [29]. Most recent post-hoc analysis of the multicenter, worldwide audit database of 9553 RBC transfused critically ill patients found that transfusions were associated with slightly lower relative-risk for in-hospital death in the most severely ill patients [30]. It is assumed that a worse outcome of patients being transfused is determined not only by the increased risk of infection, but also by non-infectious transfusion-related complications (immunomodulation, acute lung injury, circulatory overload) and RBC storage lesion [29]. These detrimental effects may be further enhanced in sepsis patients, in which sepsis-induced microcirculation impairment develops [29].

Advantages and limitations of the study. The main advantage despite the relatively small sample of study was, to our knowledge, being the very first study that included mixed a medical-surgical ICU population, not only in the biggest academical hospital of Lithuania, but also in Baltic States and Eastern Europe. The limitations of the study are as follows: this is a single-center ICU study in adults; in addition, the *A. baumannii* strains were not routinely tested for sensitivity to colistin until mid-2015, which means that some strains could be attributed to a potentially higher drug resistance category, despite their sensitivity to colistin. Additionally, we analyzed patient’s data only on ICU admission; however, patient’s condition on both the onset of VAP and on the forthcoming days could be an important prognostic factor. In order to be more precise on the results reflecting the situation in the different countries and hospitals, larger multicenter studies would be relevant. Such studies would allow for a better understanding of the impact of drug-resistant *A. baumannii* as a causative agent for VAP on in-hospital mortality.

## 5. Conclusions

Accordingly, the higher the SAPS II score, increased durations of hospital length of stay prior to ICU and prior to mechanical ventilation were related to higher resistance profiles of *A. baumannii*, timely mechanical ventilation, and ICU treatment may reduce the risk of VAP due to higher drug-resistant *A. baumannii*, especially in more severely ill patients. Carbapenem use has been shown to be associated with the risk of MDR *A. baumannii* VAP. Mortality that is due to drug-resistant *A. baumannii* VAP was high, but it has not been significantly associated with higher drug resistance of the *A. baumannii* strain. Female gender, SOFA score, and RBC transfusion were found to be independent predictors of in-hospital mortality. For instance, the estimation of red blood cells transfusion as independent predictor of in-hospital mortality allows for clinicians to be more responsible in consideration of the use RBC packs without obvious tissue hypoxia signs in purpose to preserve optimal survival. In our opinion, this investigation gives opportunities for clinicians regarding a better understanding of suspect VAP due to drug-resistant *A. baumannii* by risk factors, which are significantly associated with the drug resistance of pathogen, so as to enable easier prevention of VAP infection and mortality control. Obviously, our presented information makes it possible to predict mortality, to influence strategy of treatment, and to be able to decrease the higher costs of healthcare in this contingent of patients. We do suggest that researchers of other countries join our study for the investigation of the relation of risk factors for VAP and mortality with resistance profiles of the *A. baumannii* strain, and the estimation of the independent predictors of in-hospital mortality, which are the most important clinical issues in critical care nowadays.

## Figures and Tables

**Figure 1 medicina-55-00049-f001:**
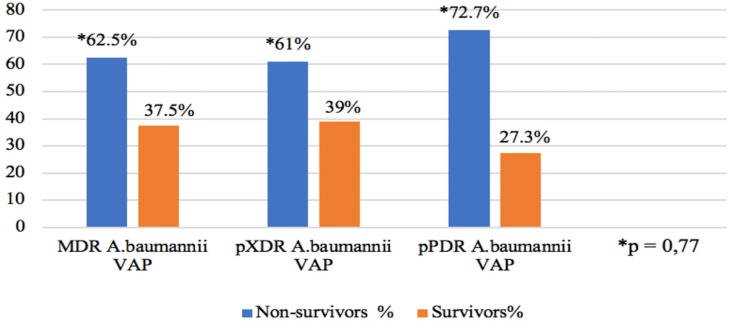
The relationship between *A. baumannii* drug resistance profile and mortality. Abbreviations: MDR—multidrug-resistant, XDR—extensively drug-resistant, pPDR—potentially pandrug-resistant, AcB—*A. baumannii*, and VAP—ventilator-associated pneumonia.

**Figure 2 medicina-55-00049-f002:**
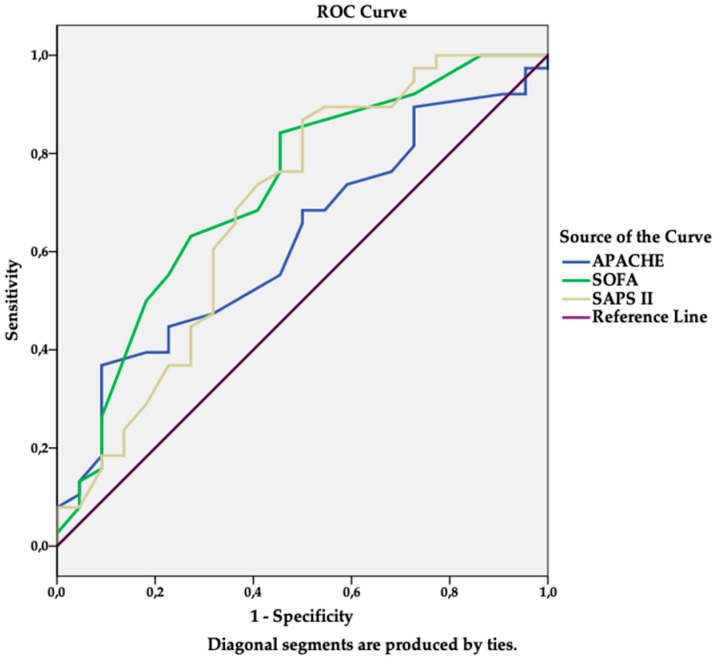
Receiver operating characteristic (ROC) curves for predicting in-hospital mortality according to Simplified Acute Physiology Score (SAPS II), Acute Physiology and Chronic Health Evaluation (APACHE) II, and Sequential Organ Failure Assessment (SOFA) scores at intensive care unit (ICU) admission.

**Table 1 medicina-55-00049-t001:** The relationship between MDR, XDR, and pPDR *A. baumannii* as pathogen of VAP and the risk factors for VAP.

Variable	All Cases *n* = 60	Drug Resistance Profiles of VAP Pathogen *A. baumannii*			*p* Value
		MDR *n* = 8	XDR *n* = 41	pPDR *n* = 11	
Gender, *n* (%)					
Female	29 (48.3)	6 (75)	18 (43.9)	5 (45.5)	0.201
Male	31 (51.7)	2 (25)	23 (56.1)	6 (54.5)	
Age (years), mean (SD)	63.95 (15)	60 (18.4)	65.37 (15.3)	61.55 (11.1)	0.555
Reintubation, *n* (%)	19 (31.7)	3 (37.5)	13 (31.7)	3 (27.3)	0.894
Surgical intervention, *n* (%)	23 (38.3)	4 (50)	16 (39)	3 (27.3)	0.593
Tracheostomy prior VAP, *n* (%)	8 (13.3)	0 (0)	5 (12.2)	3 (27.3)	0.209
Severity score on ICU admission:					
APACHE II, median (IQR)	19 (13–27.5)	18 (13.3–31.5)	18 (12-24)	23 (18–30)	0.189
SAPS II, mean (SD)	47.95 (13)	42.6 (13.8)	48.7(12.8)	49 (13.5)	0.048
SOFA, mean (SD)	11.5 (4.3)	11.8 (4.3)	11.5 (4.6)	11.36 (3)	0.981
Admission type, *n* (%)					
Surgery	17 (28.3)	2 (25)	10 (24.4)	5 (45.5)	0.186
Medical	29 (48.3)	5 (62.5)	18 (43.9)	6 (54.5)	
Trauma	14 (23.3)	1 (12.5)	13 (31.7)	0 (0)	
Coma (GCS <9), *n* (%)	8 (13.3)	2 (25.0)	4 (9.8)	2 (18.2)	0.445
Chronic disease, *n* (%)	43 (71.7)	7 (87.5)	29 (70.7)	7 (63.6)	0.508
Sepsis status on ICU admission, *n* (%)					
Sepsis	54 (90)	8 (100)	36 (87.8)	10 (90.9)	0.572
Shock	20 (33.3)	2 (25)	16 (39)	2 (18.2)	0.371
RBC transfusion prior to VAP, *n* (%)	36 (60)	4 (50)	24 (58.5)	8 (72.7)	0.573
Antibiotic treatment prior to VAP (days), median (IQR)	13 (7–18)	10.5 (5.3–13.8)	13 (7.5–20)	14 (6–19)	0.475
Antibiotic treatment prior to VAP (class), *n* (%)					
Cephalosporin	46 (76.7)	5 (62.5)	32 (78)	9 (76.7)	0.576
Penicillin ± BLI	34 (56.7)	3 (37.5)	23 (56.1)	6 (54.5)	0.626
Quinolone	6 (10)	0 (0)	5 (12.2)	1 (9.1)	0.572
Aminoglycoside	3 (5)	1 (12.5)	1 (2.4)	1 (9.1)	0.387
Carbapenem	18 (30)	4 (50)	12 (29.3)	2 (18.2)	0.036
Other	8 (13.3)	1 (12.5)	4 (9.8)	3 (27.3)	0.315
LOS (days), median (IQR)					
Hospital prior to ICU	1 (0–3.8)	0 (0–0.8)	1 (0–3.5)	2 (0–10)	0.036
ICU prior to MV	0 (0–0)	0 (0–0)	0 (0–0)	0 (0–1)	0.406
ICU prior to VAP	9 (6–13)	9.5 (5.3–13.5)	9 (6–13)	8 (5–16)	0.974
Hospital prior to MV	0 (0–2.7)	0 (0–0)	0 (0–2)	3 (0–6)	0.013
Hospital prior to VAP	11.5 (6–16.5)	10 (6–13.5)	12 (6–16)	14 (8–22)	0.424
MV prior to VAP (days), median (IQR)	8.50 (5–14)	10.5 (5–13.3)	8 (5.5–14)	7 (2–11)	0.845

SD—standard deviation, IQR—interquartile range, VAP—ventilator associated pneumonia, ICU—intensive care unit, APACHE II—Acute Physiology and Chronic Health Evaluation score II, SAPS II—Simplified Acute Physiology Score, SOFA—Sequential Organ Failure Assessment, GCS—Glasgow coma scale, RBC—red blood cells, BLI—β-lactamase inhibitors, LOS—length of stay, MV—mechanical ventilation.

**Table 2 medicina-55-00049-t002:** Univariate analysis of demographics, clinical findings and treatment factors associated with mortality of drug-resistant *A. baumannii* VAP patients.

Variable	Survivors (*n* = 22)	Non-Survivors (*n* = 38)	*p* Value
Gender, *n* (%)			
Female	6 (20.7)	23 (79.3)	**0.017**
Male	16 (51.6)	15 (48.4)	
Age (years), mean (SD)	60.77 (16)	65.79 (14.3)	0.214
Severity score on ICU admission, mean (SD)			
APACHE II	17.64 (7.1)	21.63 (9.1)	0.082
SOFA	9.41 (4.3)	12.71 (3.8)	**0.003**
SAPS II	41.73 (15.2)	51.55 (10.1)	**0.011**
Sepsis status on ICU admission, *n* (%)			
Sepsis	19 (35.2)	35 (64.8)	0.659
Shock	9 (45)	11 (55)	0.401
Chronic diseases, *n* (%)	12 (27.9)	31 (72.1)	**0.038**
Coma GCS <9, *n* (%)	4 (50)	4 (50)	0.449
Admission type, *n* (%)			
Surgery	6 (35.3)	11 (64.7)	0.981
Internal disease	11 (37.9)	18 (62.1)	
Trauma	5 (35.7)	9 (64.3)	
Reintubation, *n* (%)	8 (42.1)	11 (57.9)	0.577
Surgical intervention, *n* (%)	6 (26.1)	17 (73.9)	0.271
Tracheostomy prior to VAP, *n* (%)	1 (12.5)	7 (87.5)	0.238
RBC transfusion prior to VAP, *n* (%)	9 (25)	27 (75)	**0.03**
RBC (units), median (IQR)	0 (0–2.3)	2 (0–5)	**0.044**
LOS (days), median (IQR)			
Hospital prior to ICU	0.5 (0–3.5)	1 (0–4)	0.568
ICU prior to MV	0 (0–0.3)	0 (0–0.25)	0.876
ICU prior to VAP	8 (5.8–11)	9.5 (5.8–15)	0.442
Hospital prior to MV	0 (0–1.3)	0.5 (0–4)	0.341
Hospital prior to VAP	9.5 (6–14)	13 (7.5–20)	0.126
MV prior to VAP (days), median (IQR)	9 (5–11.5)	8 (5–15.3)	0.729
Antibiotic treatment prior to VAP (days), median (IQR)	10 (6–14.8)	14 (7.8–20)	0.164

SD—standard deviation, IQR—interquartile range, ICU—intensive care unit, APACHE II—Acute Physiology and Chronic Health Evaluation score II, SAPS II—Simplified Acute Physiology Score II, SOFA—Sequential Organ Failure Assessment score, GCS—Glasgow coma scale, RBC—red blood cells, BLI—β-lactamase inhibitors, LOS—length of stay, MV—mechanical ventilation, and VAP—ventilator associated pneumonia.

**Table 3 medicina-55-00049-t003:** Independent predictors of in-hospital mortality of drug-resistant *A. baumannii* VAP patients.

Independent Variable	Odds Ratio	95% CI	*p* Value
Gender (female)	5.26	1.21–22.83	0.027
SOFA score on ICU admission	1.28	1.06–1.53	0.008
RBC transfusion	5.97	1.41–25.27	0.015

CI—confidence interval, SOFA—Sequential Organ Failure Assessment score, ICU—intensive care unit, RBC—red blood cell transfusion.

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
