# Peer review of "Ventilator-Associated Pneumonia due to Drug-Resistant Acinetobacter baumannii: Risk Factors and Mortality Relation with Resistance Profiles, and Independent Predictors of In-Hospital Mortality"

_1010-660X, 2019, doi:10.3390/medicina55020049_

Reviewer 1 Report

Your study appears to have a somewhat high incidence of AAVAP compared to global incidences and this would be worth comments - do you know why? Do you have a seasonal variation? Are you able to say what your antibiotic policies are?

A reference that you might find usefull is - Hurley JC: World-wide variation in incidence of Acinetobacter associated ventilator associated pneumonia. A meta-regression. BMC Infect Dis 2016. https://bmcinfectdis.biomedcentral.com/articles/10.1186/s12879-016-1921-4

 The analysis is appropriate. However, the comparison within different types of MDR and XDR and pPDR reveal little, it might have been useful to compare with say Pseudomonas or some other genus.

The discussion is somewhat long. Of note, your statement '..timely mechanical ventilation and ICU treatment may reduce the risk of VAP due to higher drug-resistant A.baumannii,..' is not valid as you have not observed a causal relationship.

Figure 1 is difficult to interpret without denominator or precision data but I suspect that the low rate of survivors is not surprising and reflects the nature of the patient cohort.

Author Response

Response to Reviewer 1 Comments

Point 1: Moderate English changes required.  

Response 1: We have made some language corrections and have got approval of journal “Medicina” language editor.

Point 2: Your study appears to have a somewhat high incidence of AAVAP compared to global incidences and this would be worth comments - do you know why? Do you have a seasonal variation? Are you able to say what your antibiotic policies are?

Point 2.1.…somewhat high incidence of AAVAP compared to global incidences and this would be worth comments - do you know why?

Response 2.1The incidence of VAP remains high and steadily increasing in Lithuania. VAP is the most common hospital infection in ICU patients in our country. It accounted for almost 58% of all hospital infections in 2017. In Lithuania, we have data of the most common pathogens  of VAP (A.baumannii was the most frequent VAP pathogen until 2017; Klebsiella spp.became the leader pathogen in 2017), but we did miss the data of VAP due to multidrug-resistant bacteria. Namely VAP due to multidrug-resistant gram-negative bacteria (GNB) is the subject of our ongoing study in the 4 ICUs of the largest tertiary care university hospital in Lithuanian.  We’ve found, that multidrug-resistant A.baumannii was responsible for more than half of all VAP due to multidrug-resistant GNB cases.

In this sub-analysis we did not determinate the VAP rate, but only have analyzed the cases of disease diagnosed in mixed medical-surgical ICU within 2 years.

Point 2.2.  A reference that you might find usefull is - Hurley JC: World-wide variation in incidence of Acinetobacter associated ventilator associated pneumonia. A meta-regression. BMC Infect Dis 2016. https://bmcinfectdis.biomedcentral.com/articles/10.1186/s12879-016-1921-4

Response 2.2. Analysis of Hurley (2016) about world-wide and seasonal  variation in incidence of Acinetobacter associated VAP mentioned below was very interesting and useful. Author  has found greater than  fivefold variation of Acinetobacter associated VAP in various geographic regions. But methodology of our analysis was different from one of Hurley: the rate of VAP incidence was not the aim of our  analysis; moreover; not all cases of  Acinetobacter spp.isolated from respiratory sampling were analyzed but  only samples obtained after mechanical ventilation for at least 48 hours and from which only multidrug-resistant A.baumannii wasidentified; also, colonization and cases with gram-positive co- pathogens were excluded. 

In introduction data about VAP rate in Lithuania (2017 year) were provided from different than Hurley’s time period (up to June 2016). According to data of annual epidemiological report on surveillance of health-care associated infections in intensive-care units of  Institute of Hygiene, by the way VAP rate in Lithuania in 2016 was 23.7 cases per 1000 MV days [1], and it corresponds to those reported from Europe region (24.3 cases per 1000 MV days) in Hurley’s analysis.

To our opinion, the rate of  VAP may  increase due to prolonged life expectancy at birth, the ageing of population (declining the rates of  birth; large emigration level of young and middle age adult groups due to socio-economic reasons), what is considered to be a risk factor of greater vulnerability (due to polymorbidity and immunosuppression), and increasing demand for MV; also  - due to the inadequate funding of the health care system (due to middle income country), lacking funds for example to ensure protection of microorganisms (A.baumannii) dislocation from environment (MV machines, furniture) surfaces toward patient’s body to colonize it, later – to cause hospital infection. 

Point 2.3. Do you have a seasonal variation?

Response 2.3.We did not analyze the seasonal variability of VAP due to multidrug-resistantA.baumannii.

Point 2.4. Are you able to say what your antibiotic policies are?

Response 2.4.Our hospital has these antibiotic policies: 

·      Restrictive list of antibiotics.  

·      The guidelines for empirical antibacterial treatment of different sites infections for each department. 

·      Availability of pathogen resistance data.

·      Availability of consultations with clinical microbiologist. 

·      Review and adjustment of prescribed antibiotics every 48-72 hours.

Point 3. The analysis is appropriate. However, the comparison within different types of MDR and XDR and pPDR reveal little, it might have been useful to compare with say Pseudomonas or some other genus.

Response 3. Thank you for interesting suggestion to compare VAP due to A.baumannii with those due to P. aeruginosa. Because the purpose of my doctoral thesis is  investigation of VAP due to multidrug-resistant GNB, these comparisons are being  target of ongoing publication.

Point 4. The discussion is somewhat long. 

Response 4. Thank you for remark.  The minor shortening of discussion section were made. 

Point 5. Of note, your statement '..timely mechanical ventilation and ICU treatment may reduce the risk of VAP due to higher drug-resistant A.baumannii,..' is not valid as you have not observed a causal relationship.

Response 5. In our analysis we’ve found statistically significant associations between hospital LOS before ICU and before MV, and higher resistance profile of A.baumannii (see the data in the table 1). Although the study was carried out in the largest hospital of Lithuania, the sample size of this period  was very small; moreover, data from only one ICU were analyzed. Such sample size may be insufficient to prove the causality of this  relationship, so  the larger sample multi-center  studies of drug-resistant A.baumanniiVAP are required. 

Point 6. Figure 1 is difficult to interpret without denominator or precision data but I suspect that the low rate of survivors is not surprising and reflects the nature of the patient cohort.

Response 6. Once more, thank you for remark. Figure 1 was corrected. High rate of mortality seems to be associated with the nature of patient cohort.

Reference:

1.    Surveillance of healthcare-associated infections in intensive care units. Annual epidemiological report for 2015 [Hospitalinių infekcijų epidemiologinės priežiūros padidintos rizikos skyriuose 2015 m. ataskaita]. Available from: http://www.hi.lt/uploads/pdf/hospitalines/duomenu ataskaitos/RITS-2016 m.pdf. [Internet; cited 20.01.19].

Reviewer 2 Report

That's an interesting study and results as well.

A few corrections:

line 18: delete "backgraond" what's wrettin is just the objective.

line 20: delete "materials"

line 59: state what "MV" days stands for.

The introduction can be improved by adding more about the history of VAP and what antibiotics uses to treat it.

Try to improve the quality of Fig 2. 

The reason of admission (line 87), I found the word "medical" is vague, can you specify more? Do you have the data? It would be informative to add such data.

Overall, I like the discussion section (even though it's too long) and the mention of the limitations and the end.

Regards,

Author Response

Response to Reviewer 2 Comments

Point 1: English language and style are fine/minor spell check required.  

Response 1: We have made some language corrections and have got approval of journal “Medicina” language editor.

Point 2:line 18: delete "backgraond" what's wrettin is just the objective.

Thank you for your remark about background. We’ve made same corrections in our abstract  and added background.

Point 3.line 20: delete "materials"

Response 3.Word “material” has been deleted.

Point 4.line 59: state what "MV" days stands for.

MV is abbreviation of mechanical ventilation. Number of cases per 1000 mechanical ventilation days is one of the indices of VAP rate. We made correction and it was written in full words.

Point 5.The introduction can be improved by adding more about the history of VAP and what antibiotics uses to treat it.

Response 5. Thank you for suggestion. Some facts about history of VAP due to A.baumanniiand its antibacterial treatments have been  added.

Point 6.Try to improve the quality of Fig 2. 

Response 6.The quality of Fig 2 has been improved.

Point 7. The reason of admission (line 87), I found the word "medical" is vague, can you specify more? Do you have the data? It would be informative to add such data.

Response 7. The meaning of “hospitalization for medical reason” means hospitalization due to internal diseases. As you recommended, more detailed data on proportion of miscellaneous internal disease of A.baumanniiVAP patients was added to the “Results” section.

Point 8.Overall, I like the discussion section (even though it's too long) and the mention of the limitations and the end.

Response 8.With respect to the last remark minor reduction in discussion section was accomplished.